# GuessBench: Sensemaking Multimodal Creativity in the Wild

## Abstract

We propose GuessBench, a novel benchmark that evaluates Vision Language Models (VLMs) on modeling the pervasive, noisy, and pluralistic human creativity. GuessBench sources data from "Guess the Build", an online multiplayer Minecraft minigame where one player constructs a Minecraft build given a concept (e.g., caterpillar) and others try to guess it with natural language hints, presenting a valuable testbed for sensemaking creativity in the wild with VLMs acting as guessers. We curate 1500 images from the actual gameplay and design 2000 problems spanning static and dynamic image settings, natural language hints of varying completeness, and more. Extensive experiments with six open/API VLMs and five reasoning enhancement approaches demonstrate that GuessBench presents a uniquely challenging task in creativity modeling: even the start-of-the-art GPT-4o is incorrect on 34% of instances, while we observe a huge performance gap (13.87% vs. 53.93% on average) between open and API models. When used as a resource to improve VLMs, fine-tuning on the reasoning traces for GuessBench problems improves visual perception tasks by 15.36% on average. Further analysis reveals that VLM performance in creativity sensemaking correlates with the frequency of the concept in training data, while the accuracy drops sharply for concepts in underrepresented cultural contexts and low-resource languages.

## 1 Introduction

Vision Language Models (VLMs) have demonstrated remarkable capabilities across perception, knowledge, and reasoning problems (OpenAI, 2024; Wang et al., 2024d; Yao et al., 2024; Bai et al., 2025). From objective tasks to subjective contexts, recent works have explored creative tasks for generative models, evaluating their skills in creative generation and problem-solving across language (Wang et al., 2024c; Lv et al., 2024; Shen & Guestrin, 2025; Minh et al., 2025), image (Lifshitz et al., 2023; Ng et al., 2024; Fang et al., 2025; White et al., 2025; Han et al., 2025), audio (Cherep et al., 2024), and video modalities (Miller et al., 2024; Han et al., 2024; Feng et al., 2024b; Jiang et al., 2024; Wang et al., 2024a).

We identify two key gaps in the research of creative generative modeling. 1) *Creativity in the wild*: while existing research holds VLMs to high artistic standards (Wang et al., 2023a; Chakrabarty et al., 2024; Tang et al., 2024; Zhang et al., 2024), it struggles to incorporate the imperfect creativity of diverse VLM users. Their creativity is *pervasive*, as many of their VLM requests (e.g., polishing a photo and generating a flowchart) require creative decision making (Kim et al., 2025a); their creativity is *noisy*, as the average VLM user is not artistically trained and did not memorize the elite art in VLM training data (Barton, 2013); their creativity is *pluralistic*, as different individuals could have varying interpretations for the same entity and concept (Sorensen et al., 2024; Feng et al., 2024c). As such, reflecting and modeling creativity *in the wild* is a crucial step in aligning the creative capabilities of VLMs with diverse VLM users. 2) *Sensemaking creativity*: while most research focuses on *generative* creativity where models are tasked with generating engaging text or pretty images (Lu et al., 2024c; Gokaslan et al., 2024; Senft-grupp et al., 2025; Peng et al., 2025), there is limited exploration on model capabilities in analyzing and decoding creative constructs. Quantifying and augmenting VLMs' skills in understanding and *sensemaking* creativity could assist art teaching, build a reward model for creativity, and more.

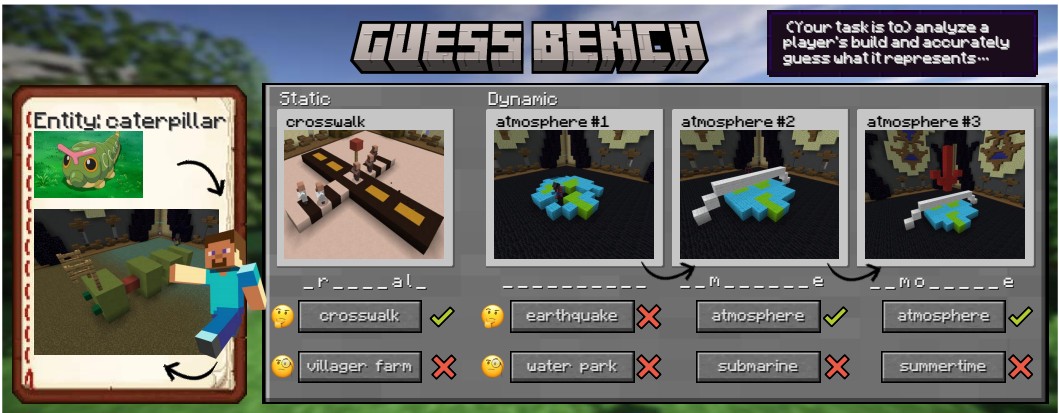

Figure 1: GUESSBENCH contains player-built Minecraft constructions representing real-world entities and concepts. VLMs are required to infer the represented concept from image(s) and hint(s). Two settings are offered: static (one image and hint) and dynamic (three sequences of progressively refined builds and hints; a correct answer in any attempt marks all subsequent attempts correct).

To this end, we propose GUESSBENCH, a creativity understanding dataset where VLMs guess and work out the underlying theme of a Minecraft image (Figure 1). GUESSBENCH is sourced from "Guess the Build", a minigame on the Hypixel Minecraft multiplayer network[1]: Given a concept (e.g., caterpillar or oasis), one player creates a Minecraft build conveying the concept, while other players try to guess the concept with optional natural language hints. We curate 1500 images from the actual gameplay and design 2000 problems under two evaluation settings: *static*, where the VLM is only presented with the completed Minecraft build for one attempt, and *dynamic*, where the VLM is presented with a sequence of images where the build is increasingly complete for multiple attempts. We posit that GUESSBENCH presents a valuable *creativity in-the-wild* setting where data is sourced from a diverse and global player community with a spectrum of language and cultural backgrounds.

We evaluate a wide range of state-of-the-art VLMs and enhancement approaches (e.g., Self-Refine (Madaan et al., 2023) and Image Retrieval (Zhu et al., 2024)) on GUESSBENCH. We find that *creativity sensemaking* is challenging, even the state-of-the-art GPT-4o only achieves 57.8% and 66.0% in static and dynamic settings. GUESSBENCH also introduces a quantitatively novel (Li et al., 2025a) dataset: compared to various VQA datasets (Lu et al., 2024b; Guan et al., 2024), GUESSBENCH is more challenging, better separates model performance, and reveals novel insights about model strengths and weaknesses (Figure 2). GUESSBENCH could also be a useful training resource: by fine-tuning VLMs on the reasoning traces in GUESSBENCH, their performance on visual datasets improves by 15.36% on average. Further analysis reveals that VLM creativity sensemaking degrades substantially for low-resource languages, correlates with the frequency of the concept in training data, and suffers from sycophancy when facing contradictory user requests.

## 2 GUESSBENCH

We propose GUESSBENCH, a novel and challenging benchmark designed to evaluate the creative understanding capabilities of VLMs. GUESSBENCH is based on the "Guess the Build" Minecraft minigame, where players construct a Minecraft build given a concept (e.g. caterpillar) and others try to guess it with natural language hints. We first introduce two problem settings for VLMs acting as a guesser, *static* and *dynamic*, where the Minecraft build image either stays fixed or gradually becomes more complete as the player develops their build (§2.1). To ensure the reasonableness, diversity, and difficulty of the problems, we collect images from real Minecraft gameplay scenarios and design well-crafted natural language hints with varying difficulty levels (§2.2). We then describe the evaluation metrics in §2.3 and present dataset statistics in Table 1.

---

[1]https://hypixel.net

## 2.1 TASK SETTING

**Static Task Setting.** Given an image of a Minecraft build $b$ and a corresponding natural language hint $h$, the VLM's goal is to decipher player creativity and identify what concept does the build represent. This task can be formally defined as: $\text{VLM}(b, h) = c$, where $\text{VLM}(\cdot)$ denotes the VLM, and $c$ is the textual output representing the VLM's guessed concept (e.g. oasis or caterpillar).

**Dynamic Task Setting.** Consider a sequence of Minecraft build images denoted by $\boldsymbol{b}_T = (b_1, b_2, \ldots, b_T)$, where $b_t$ represents the build image at time step $t$. Correspondingly, we define a sequence of hints as $\boldsymbol{h}_T = (h_1, h_2, \ldots, h_T)$, and a sequence of the VLM's previous guesses as $\boldsymbol{c}_{T-1} = (c_1, c_2, \ldots, c_{T-1})$, where $\boldsymbol{c_0}$ is set as an empty sequence to indicate no previous guesses at the start. In our proposed GUESSBENCH, both the Minecraft build images and natural language hints become progressively more complete through time, presenting a temporal and dynamic setting where VLMs need to work with incomplete information. Based on this setup, the task can be formally defined as: $\text{VLM}(\boldsymbol{b}_T, \boldsymbol{h}_T, \boldsymbol{c}_{T-1}) = \boldsymbol{c}_T, T \geq 1$.

## 2.2 DATA CURATION

**Build Collection.** To collect representative visual data, we manually participate in the "Guess the Build" online multiplayer game on the Hypixel server and capture screenshots of the constructed builds. During the image selection process, we also remove low-quality builds such as those where players spell out the answer using blocks instead of building its visual representation, or where no meaningful construction is present (e.g., merely holding an apple when the target word is "apple"). In total, GUESSBENCH comprises of 500 carefully curated build sets. For the *dynamic* task, we set the number of attempts $T = 3$, collecting three successive Minecraft build images for each set. For the *static* task, we use the third-attempt image from the dynamic task (i.e., $b_3$) as the representative build for each set. This results in a total of 1500 images featuring diverse concepts and varying levels of completeness.

| Statistics | Value |
|---|---|
| Minecraft build sets | 500 |
| *Answer* | |
| - Unique answers/tokens | 424/678 |
| - Maximum/Average length | 4/2.1 |
| **Static Task** | |
| *Minecraft Build Images* | |
| - Total Images | 500 |
| - Images per set | 1 |
| - Average size (px) | $1188 \times 753$ |
| - Maximum size (px) | $1920 \times 1080$ |
| *Hints* | |
| - Unique hints/tokens | 481/120 |
| - Maximum/Average length | 127/78.3 |
| **Dynamic Task** | |
| *Minecraft Build Images* | |
| - Total images | 1500 |
| - Images per set | 3 |
| - Average size (px) | $1150 \times 721$ |
| - Maximum size (px) | $1920 \times 1080$ |
| *Hints* | |
| - Unique hints/tokens | 933/125 |
| - Maximum/Average length | 127/61.2 |

Table 1: Statistics of GUESSBENCH. The unique tokens and lengths of hints and answers are measured using the GPT-4o tokenizer.

**Hint Generation.** To more faithfully simulate the complete gameplay of "Guess the Build", we draw inspiration from its progressive hint-revealing mechanism and design the corresponding hint for each guessing attempt. Specifically, suppose the ground-truth answer consists of $N$ letters (e.g., caterpillar). For the **dynamic task**, the first-attempt hint discloses the number of words in the answer and the number of letters in each word (e.g., _ _ _ _ _ _ _ _ _ _ _). In the second attempt, based on the previous hint, we additionally reveal $\lceil N/8 \rceil$ randomly selected letters (e.g., _ _ _ _ r _ i _ _ _ _). The third-attempt hint builds on the second attempt by further providing an additional $\lceil N/4 \rceil - \lceil N/8 \rceil$ randomly chosen letters (e.g., c _ _ _ r _ i _ _ _ _). For the **static task**, we directly adopt the third-attempt hint used in the dynamic task as the sole hint for the question.

In addition to the symbolic representation, we also provide the hints in natural language to aid VLMs in better interpreting the partial information (e.g., The answer consists of 1 word. The 1st word has 11 letters. The 5th letter is 'r'. The 7th letter is 'i'.). Two illustrative examples for both the dynamic and static tasks are presented in Appendix F.1.

The choice of the divisors 8 and 4 serves two purposes: (1) to better mimic the hint progression in the original "Guess the Build" game, and (2) to balance the difficulty across different answer lengths, thereby enabling a more nuanced evaluation of model performance on the GUESSBENCH.

## 2.3 EVALUATION METRICS

To evaluate the responses generated by VLMs, we first follow the methodology proposed in Math-Vista (Lu et al., 2024b) by employing GPT-4o to extract the predicted guess from each response (Appendix A.2), and subsequently adopt accuracy as the evaluation metric. We observe that the median and mean answer lengths across all build sets are 8 and 8.15 letters respectively. Motivated by this observation, we categorize questions with answers of length less than or equal to 8 as *short* answer questions, and those with longer answers as *long* answer questions. In addition to reporting the overall accuracy, we also report the accuracy separately for the short and long subsets, aiming to investigate whether the length of the concept word is an impact factor.

Specifically, for the three-attempt dynamic tasks, if a VLM correctly answers a build-set question in either the first or second attempt, it is exempted from answering the question again in subsequent attempts, which are automatically marked as correct.

## 2.4 ETHICAL CONSIDERATIONS

We take various steps to ensure the ethical compliance of GUESSBENCH. We first inspect the Minecraft terms of service[2] for multiplayer servers as well as the Hypixel server's terms of service[3], ensuring that players consent to be viewed by a larger audience when joining multiplayer servers and the academic use and anonymized resharing of in-game content is within intended use. To protect player privacy, we apply blurring to any visible player IDs for anonymization. We manually inspect all collected images and remove any that feature hateful or offensive builds.

# 3 EXPERIMENT SETTINGS

## 3.1 MODELS AND IMPLEMENTATION

We evaluate nine widely used VLMs on GUESSBENCH, including three API and six open models. The API models include GPT-4o (OpenAI, 2024), GPT-4o-mini (OpenAI, 2024), and Gemini-2.0-Flash (Team et al., 2023). For the open models, we evaluate Gemma-3-27B (Team et al., 2025), InternVL2.5-78B-MPO (Wang et al., 2024d), InternVL2.5-8B-MPO (Wang et al., 2024d), Qwen2.5VL-72B (Bai et al., 2025), Qwen2.5-VL-7B (Bai et al., 2025), and MiniCPM-V2.6 (Yao et al., 2024). For reference, we additionally report baselines of human performance, image-only input, and text-only input. We also conduct human evaluation following the same evaluation procedure as applied to the VLMs.

To ensure the reproducibility, we standardize the decoding configurations across all VLMs by setting the temperature to 0.0 and top_p to 1.0, or by disabling sampling via setting do_sample to False. The specific prompts and hyperparameters used for each VLM are detailed in Appendix A. All experiments are conducted using eight NVIDIA A100-SXM4-40GB GPUs.

## 3.2 REASONING APPROACHES

The models we tested employ chain-of-thought (CoT) reasoning by default, while we further explore several alternative reasoning strategies specifically for the strongest model GPT-4o. The results are reported in Table 2, and the corresponding instructions for each approach are provided in the Appendix A.3. In the **w/o CoT** configuration, the prompt explicitly instructs the model to generate a direct answer without any intermediate reasoning steps. We explore this setup to investigate whether the intermediate reasoning steps are helpful in the task of creativity sensemaking. In the **One-shot** setting, we provide a single demonstration. For *static* tasks, this demonstration includes an image of a Minecraft build, a hint, and the corresponding answer. For *dynamic* tasks, the demonstration consists of the current attempt along with all previous attempts, each accompanied by its associated image, hint, and answer. The **Self-Consistency** (Wang et al., 2023b) generates three independent responses and adopts the final prediction through majority voting. If all three responses differ, the last guess is selected as the final answer. The **Self-Refine** (Madaan et al., 2023) approach prompts

---

[2]https://www.minecraft.net/en-us/terms/r2
[3]https://hypixel.net/terms

| Model | Static | | | Dynamic #1 | | | Dynamic #2 | | | Dynamic #3 | | |
|---|---|---|---|---|---|---|---|---|---|---|---|---|
| | Short | Long | All | Short | Long | All | Short | Long | All | Short | Long | All |
| Human | **78.3** | **80.6** | **79.6** | **24.3** | **17.2** | **21.2** | **56.8** | **58.6** | **57.6** | **83.8** | **89.7** | **86.4** |
| Image Only | 23.2 | 22.0 | 22.6 | 6.9 | 6.2 | 6.6 | 17.0 | 14.5 | 15.8 | 30.1 | 26.6 | 28.4 |
| Text Only | 17.0 | 18.3 | 17.6 | 0.4 | 1.7 | 1.0 | 7.7 | 12.0 | 9.8 | 20.8 | 28.2 | 24.4 |
| **API Models** | | | | | | | | | | | | |
| GPT-4o | **58.7** | **56.8** | **57.8** | **10.8** | **12.0** | **11.4** | **40.2** | **44.4** | **42.2** | **65.6** | **66.4** | **66.0** |
| Gemini-2.0-Flash | 47.5 | 44.0 | 45.8 | 8.5 | 7.5 | 8.0 | 29.3 | 25.7 | 27.6 | 52.9 | 50.6 | 51.8 |
| GPT-4o-mini | 40.9 | 32.4 | 36.8 | 7.3 | 7.1 | 7.2 | 29.0 | 23.7 | 26.4 | 49.8 | 37.8 | 44.0 |
| **Open Models** | | | | | | | | | | | | |
| Gemma-3-27B | **30.5** | **24.1** | **27.4** | **5.8** | **4.6** | **5.2** | **15.4** | **14.9** | **15.2** | **34.4** | **25.7** | **30.2** |
| InternVL2.5-78B-MPO | 25.5 | 18.7 | 22.2 | 3.1 | 2.1 | 2.6 | 12.0 | 7.9 | 10.0 | 23.9 | 14.1 | 19.2 |
| Qwen2.5VL-72B | 15.4 | 8.3 | 12.0 | 2.3 | 0.8 | 1.6 | 8.5 | 6.2 | 7.4 | 18.1 | 9.5 | 14.0 |
| MiniCPM-V2.6 | 12.4 | 6.2 | 9.4 | 1.9 | 1.2 | 1.6 | 4.6 | 3.3 | 4.0 | 10.8 | 5.8 | 8.4 |
| Qwen2.5-VL-7B | 9.3 | 5.4 | 7.4 | 1.2 | 1.2 | 1.2 | 6.6 | 2.5 | 4.6 | 7.3 | 4.1 | 5.8 |
| InternVL2.5-8B-MPO | 8.9 | 2.9 | 6.0 | 1.9 | 1.7 | 1.8 | 4.6 | 2.5 | 3.6 | 6.9 | 2.9 | 5.0 |
| **Augmented GPT-4o** | | | | | | | | | | | | |
| w/o CoT | 59.1 | **58.1** | 58.6 | **11.6** | 12.0 | 11.8 | 42.1 | 44.4 | 43.2 | 68.0 | **67.6** | 67.8 |
| One-shot | 59.8 | 53.1 | 56.6 | 10.4 | 11.2 | 10.8 | 42.5 | 42.7 | 42.6 | **70.7** | 64.7 | 67.8 |
| Self-Consistency | 59.1 | 55.2 | 57.2 | 11.2 | **13.3** | **12.2** | 39.4 | 42.3 | 40.8 | **70.7** | 65.6 | 68.2 |
| Self-Refine | **61.0** | **58.1** | **59.6** | 9.3 | 10.4 | 9.8 | **43.2** | **46.1** | **44.6** | 70.3 | **67.6** | **69.0** |
| Image Retrieval | 58.3 | 53.9 | 56.2 | 9.3 | 12.0 | 10.6 | 42.1 | 42.3 | 42.2 | 67.6 | 65.6 | 66.6 |

Table 2: Evaluation results on GUESSBENCH. **Bold** values indicate the best performance in each category. **All** denotes the overall accuracy for each task. The results demonstrate the challenging nature of GUESSBENCH and reveal a clear performance disparity between open and API models.

the model to evaluate its own response. If it deems the initial prediction correct, it outputs the answer directly. Otherwise, it reconsiders its response up to two additional times. If the model is still unable to affirm the correctness of any prediction, the last guess is adopted as the final answer. Building on this, and tailored to the unique characteristics of our task, we further propose an extension of the Self-Refine method that incorporates **Image Retrieval** (Zhu et al., 2024) to better support the model's self-evaluation. During self-evaluation, the model uses its current guess as a query to retrieve the top image result from Google Images[4]. This retrieved image is then incorporated into the model's reasoning to reassess the validity of its prediction.

## 4 RESULTS

Table 2 presents the performance of six open and API models on GUESSBENCH, from which we draw several key observations.

**First, GUESSBENCH is highly challenging.** The best-performing model, GPT-4o, attains only 57.8% accuracy on static tasks and 66.0% accuracy on the final attempt of dynamic tasks. This indicates that in at least 34.0% of cases, even the strongest VLM fails to interpret user creativity. Moreover, this performance still lags far behind human capability: human evaluation surpasses GPT-4o by 37.7% on static tasks and by 30.9% on dynamic tasks.

**Second, we observe a significant performance gap between open and API models.** GPT-4o outperforms the best open-source model by 111% in static tasks and by 119% in the third attempt of dynamic tasks. This disparity indicates that, although open models demonstrate competitive or even superior performance to API models on earlier benchmarks (Bai et al., 2025; Wang et al., 2024d), their performance degrades substantially on GUESSBENCH's novel tasks. This suggests that previous benchmark success may stem from memorization of similar data and tasks, rather than a genuine and overall improvement of visual perception and reasoning. Consequently, open models exhibit limited creative and conceptual understanding.

**Third, API models benefit considerably from the iterative refinement in the dynamic setting.** In contrast, open models show minimal improvement, and in some cases, performance even declines. For the third attempt in the dynamic task, models are additionally provided with less complete Minecraft builds and imperfect hints from the previous two attempts. Under this setting, the average accuracy of API models increases by 15.24% compared to their static task performance. However,

---

[4]https://www.google.com/imghp.

open models exhibit almost no improvement, and the performance of InternVL2.5-78B-MPO even decreases by 13.51%. This indicates that open models struggle to understand the incomplete builds in GUESSBENCH, further revealing their limited ability to generalize to unfamiliar tasks.

**Fourth, none of the reasoning strategies tested leads to substantial performance improvements.** Among them, Self-Refine proves to be most effective, while Image Retrieval performs poorly. Self-Refine increases accuracy by only 3.11% on static tasks and 4.55% on the third attempt of dynamic tasks. This outcome suggests that the challenges posed by GUESSBENCH cannot be adequately addressed through existing strategies to enhance VLMs. Interestingly, Image Retrieval causes a 2.77% decrease in static task accuracy and only yields a 0.91% increase in dynamic task attempt 3. Our in-depth case study reveals that some questions in GUESSBENCH have multiple valid answers, each corresponding to a different visual representation. When a model's initial guess is correct, the retrieved image may reflect an alternative but still valid construction. However, due to differences between the original and retrieved images, the model may become confused and abandon its correct initial guess (Appendix F.2). This highlights a current limitation in the model's ability to handle many-to-one mappings between images and textual answers, particularly in scenarios requiring creative visual perception. It also indicates that while most existing VLM enhancement methods on evaluated on common domains such as VQA and math, progress might not be generalizable to the wide spectrum of other VLM uses such as comprehending user creativity.

**Lastly, GUESSBENCH is fundamentally a cross-modal benchmark, where relying solely on text or image input is insufficient.** On static tasks, GPT-4o achieves only 22.6% accuracy when given image-only input and 17.6% accuracy with text-only input. These are both substantially lower than the 57.8% accuracy achieved with multimodal input. A similar pattern is observed in dynamic tasks. This demonstrates that a single modality alone is inadequate for solving the tasks in GUESSBENCH, highlighting the necessity of effective multimodal integration for successful task completion.

## 5 ANALYSIS

### 5.1 QUANTITATIVE EVALUATION OF GUESSBENCH VIA AUTOBENCHER METRICS

We adopt three evaluation metrics, Difficulty, Separability, and Novelty, from the Auto-Bencher (Li et al., 2025a) framework to quantitatively assess the quality of our dataset. The **Difficulty** of a benchmark is defined as the lowest error rate achieved by any model, reflecting the overall challenge posed by the task. **Separability** measures the degree to which models can be distinguished based on their performance, calculated as the mean absolute deviation of accuracies across models. **Novelty** quantifies how distinct the benchmark is from

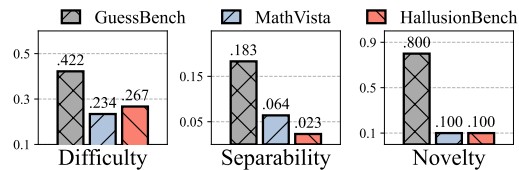

Figure 2: The AutoBencher scores of GUESS-BENCH, MathVista, and HallusionBench, higher is better. GUESSBENCH demonstrates higher levels of difficulty, separability, and novelty compared to the other two benchmarks.

existing ones, computed as one minus the rank correlation between the accuracy vector on the current dataset and the most similar predicted accuracy vector derived from existing datasets. These metrics provide a quantitative measurement of dataset and task quality.

To compute these metrics, we evaluate five vision-language models: two API models (GPT-4o and Gemini-2.0-Flash) and three open models (InternVL2.5-78B-MPO, Qwen2.5VL-72B, and MiniCPM-V2.6). We benchmark GUESSBENCH alongside MathVista (Lu et al., 2024b) and HallusionBench (Guan et al., 2024), with the results summarized in Figure 2. The detailed accuracy of all models on all involved benchmarks is presented in Appendix A.6.

**Difficulty.** GUESSBENCH achieves a difficulty score of 0.422, which is 80.34% higher than Math-Vista and 58.05% higher than HallusionBench. This significant increase indicates the heightened complexity and challenge posed by GUESSBENCH. **Separability.** GUESSBENCH obtains a separability score of 0.183, which is 2.86 times that of MathVista and 7.96 times that of HallusionBench. This suggests that GUESSBENCH is substantially more effective at differentiating the performance of various models in understanding creativity in the wild, revealing the strengths and weaknesses of models that were unclear with previous datasets. **Novelty.** We assess novelty by considering MM-

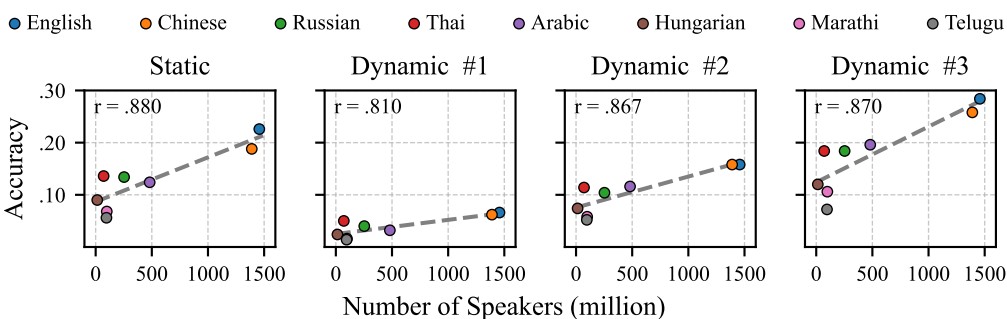

Figure 4: GPT-4o performance across eight languages with different resource availability. Performance declines for low-resource languages across four settings and shows a strong positive correlation with language prevalence, measured by Pearson correlation r.

Bench V1.1 Test (EN) (Liu et al., 2024b) and AI2D (Kembhavi et al., 2016) as prior datasets. Based on this metric, GUESSBENCH achieves higher novelty score than both MathVista and Hallucination-Bench. Moreover, the model-specific accuracy scores on GUESSBENCH reveal a clear performance gap between open-source and proprietary models in addressing creative reasoning tasks. This further underscores the novelty and significance of our dataset and benchmark task in evaluating real-world multimodal understanding.

## 5.2 IMPACT OF GUESSBENCH FINE-TUNING ON RELATED TASKS

To investigate whether creativity sensemaking benefits VLMs on other tasks, we conduct a series of transfer learning experiments using the Qwen2-VL-7B. We evaluate performance under three distinct settings: without tuning, synthetic tuning, and mixed tuning, across three benchmarks: MathVista, MultiChartQA (Zhu et al., 2025), and BLINK (Fu et al., 2024). Detailed results are shown in Figure 3. In the synthetic tuning setting, we use 289 instances correctly answered by GPT-4o from GUESS-

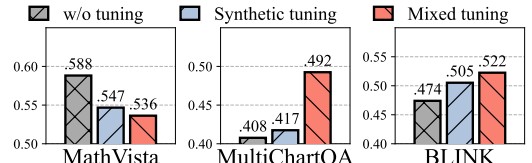

Figure 3: The performance of various fine-tuning methods. Fine-tuning improves performance on both MultiChartQA and BLINK, whereas it reduces performance on MathVista.

BENCH to fine-tune Qwen2-VL-7B. In the mixed tuning setting, we augment this with 289 additional examples correctly answered by GPT-4o from target benchmarks, totaling 578 training samples. For evaluation, training data are excluded from benchmarks, and 289 held-out examples per benchmark are randomly selected as test sets. Fine-tuning hyperparameters are identical across settings (Appendix A.5). Our findings show that synthetic tuning leads to noticeable performance improvements on MultiChartQA and BLINK, although performance on MathVista declines. Under mixed tuning, accuracy improves further on MultiChartQA and BLINK, with particularly substantial gains on MultiChartQA, while MathVista performance continues to decline.

Since GUESSBENCH emphasizes perceptual and interpretive skills, fine-tuning on it boosts performance on perception-driven tasks such as BLINK and, to a lesser extent, MultiChartQA, but can be detrimental to tasks requiring precise logical or mathematical reasoning, such as MathVista. Although both synthetic and mixed tuning promote CoT reasoning in Qwen2-VL-7B, these gains are mostly limited to tasks demanding high-level visual understanding and flexible sensemaking. Overall, GUESSBENCH enhances interpretive competence for tasks that require nuanced perception and contextual inference.

## 5.3 EVALUATING MULTILINGUAL ROBUSTNESS IN CREATIVE TASKS

To investigate whether VLMs exhibit performance disparities across different linguistic environments, we conduct a multilingual evaluation based on the tasks categorized as Image Only in Table 2. For each selected task, we provide identical prompts translated into various languages and

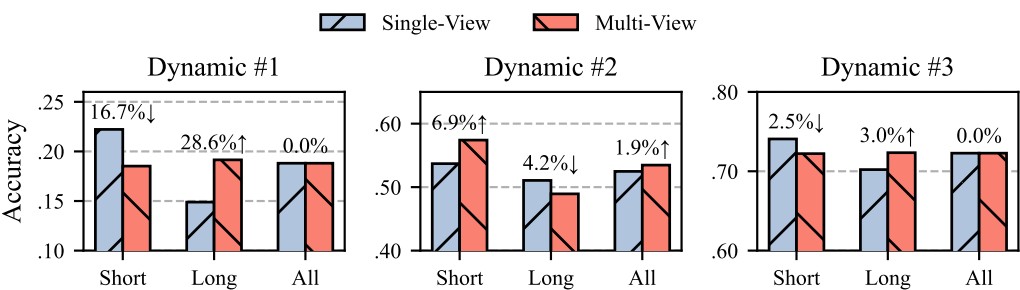

Figure 6: GPT-4o performance under Single-View and Multi-View settings. Multi-View Minecraft build images initially affect performance on both short- and long-answer questions in the Dynamic task, but the impact diminishes with more attempts, and overall performance stays consistent.

translate the corresponding answers into the same target languages. GPT-4o is then tasked with reasoning under different language settings of the same problem.

To quantify language prevalence, we refer to the number of speakers (in millions) for each language as reported by Wikipedia, and visualize the relationship in Figure 4. We observe that for high-resource languages such as English and Chinese, GPT-4o achieves significantly higher accuracy across all three attempts on both static and dynamic tasks, compared to languages with fewer speakers. Across all tasks, the Pearson correlation coefficient between the number of speakers and GPT-4o's accuracy is at least 0.810, indicating a strong positive correlation. These findings suggest that the creative sensemaking capabilities of VLMs are substantially reduced in low-resource language settings, revealing a critical gap in current multilingual generalization and raising concerns about the equity of model performance when serving diverse language speakers.

### 5.4 IMPACT OF CONCEPT FREQUENCY

To investigate the impact of concept frequency on model performance, we adopt the Infinigram (Liu et al., 2024a) approach and use Dolma-v1.7 (2.6T tokens) as our corpus to obtain concept frequency statistics for each problem. We then evaluate the accuracy of GPT-4o under the static task setting. The accuracy for each bin represents the average accuracy of all samples within that range. As shown in Figure 5, we visualize the results using two methods: on the left, the x-axis represents the top-k percentage of answers sorted in ascending order of frequency; on the right, the x-axis directly reflects the raw word frequency in ascending order. We compute the Pearson correlation coefficient between accuracy and average word frequency across bins, yielding values of 0.852 and 0.770 respectively, indicating a strong positive correlation. These results suggest that when GPT-4o attempts to understand creativity in the wild, it is more likely to succeed with higher-frequency concepts, revealing limitations in its sensemaking capabilities for long-tail entities and concepts underrepresented in the training data.

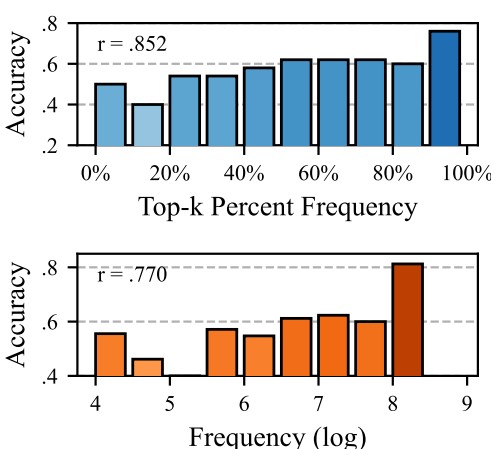

Figure 5: GPT-4o performance by concept frequency. Bins with fewer than three answers are omitted. Darker colors indicate higher accuracy. Both plots show a strong positive correlation between accuracy and concept frequency, measured by Pearson correlation r.

### 5.5 EFFECT OF MULTI-VIEW INPUTS

To explore whether providing VLMs with additional physical views of the same Minecraft build enhances their creative sensemaking, we evaluate GPT-4o under the Dynamic task setting. Specifi-

cally, we sample 101 sets of Minecraft build images, and for each attempt within the Dynamic task, we supplement the original input with two additional images captured from different viewpoints of the same build. As shown in Figure 6, an intriguing pattern emerges: when analyzing short-answer and long-answer questions individually, additional viewpoints have some influence on GPT-4o's accuracy, but this influence decreases markedly with more attempts. In contrast, when considering overall accuracy across both question types, the impact of additional viewpoints becomes minimal or even negligible.

These findings suggest that merely offering more visual angles of a build does not substantially aid in identifying what the build represents, echoing VLMs' limitations in multi-view reasoning (Zhang et al., 2025). This further highlights the challenging nature of GUESSBENCH, as it requires VLMs to engage in deeper creative understanding, association, and reasoning rather than relying solely on expanded visual input.

## 6 RELATED WORK

**VLM Evaluation.** VLMs are increasingly popular and demonstrate great performance across a wide range of vision-language tasks. As such, evaluating their capabilities has emerged as a critical research problem. Existing benchmarks predominantly focus on objective tasks, including: (1) General Question Answering, such as MMBench (Liu et al., 2024b), SEED-Bench (Li et al., 2024a), and MMMU (Yue et al., 2024); (2) Optical Character Recognition (OCR), such as OCR-VQA (Mishra et al., 2019) and OCRBench (Liu et al., 2024c); (3) Graphic Understanding, including AI2D (Kembhavi et al., 2016), CharXiv (Wang et al., 2024e), and MultiChartQA (Zhu et al., 2025); and (4) Mathematics, such as MathVista (Lu et al., 2024b), MathVision (Wang et al., 2024b), and MathV360K (Shi et al., 2024). However, current VLM benchmarks fall short in systematically exploring the *subjective* capabilities of these models. In this work, we propose to evaluate VLMs through tasks that assess understanding and sensemaking creativity through GUESSBENCH, offering a lens into the models' performance on more subjective dimensions.

**Creativity in Generative Models.** Prior work on the creativity of generative models has primarily focused on elite-level creativity, such as high-quality writing (Chakrabarty et al., 2024; Tian et al., 2024a; Lu et al., 2024c; Lee et al., 2023; Kim et al., 2025b; Gómez-Rodríguez & Williams, 2023; Chen & Ding, 2023; Bae & Kim, 2024; Marco et al., 2024; Chen et al., 2024), scientific discovery (Kang et al., 2022; Hope et al., 2022), image synthesis (Huang et al., 2023; Shah et al., 2025; Kamb & Ganguli, 2024; Isajanyan et al., 2024; Feng et al., 2024a; Lu et al., 2024a), problem solving (Tian et al., 2024b; Anonymous, 2025; Alavi Naeini et al., 2023; G Leon et al., 2024; Nair et al., 2024), code generation (Lu et al., 2025; Kranti et al., 2024), combinational creativity (Zhong et al., 2024; Peng et al., 2025; Nagarajan et al., 2025; Favero et al., 2025; Li et al., 2024c), and embodied reasoning (Park et al., 2024; White et al., 2025; Dong et al., 2024; Jia et al., 2025; Chaturvedi et al., 2024; Li et al., 2024b; Qin et al., 2024; Li et al., 2025b). However, prior research has largely overlooked creativity among the general public. Such widespread and personalized creativity, known as "creativity in the wild", is often imperfect. To address this, we propose GUESSBENCH to measure VLMs' understanding of creativity in the wild.

## 7 CONCLUSION

We present GUESSBENCH, a novel benchmark designed to evaluate VLMs' capacity for creativity sensemaking in the wild. By leveraging gameplay data from the Minecraft minigame "Guess the Build", our dataset captures the pervasive, noisy, and pluralistic nature of real-world human creativity, offering a unique and challenging testbed for VLMs acting as guessers. Through comprehensive experiments, we demonstrate that state-of-the-art models such as GPT-4o struggle with dynamic reasoning and decoding linguistically/culturally underrepresented contexts. Our findings underscore the limitations of existing VLMs in modeling diverse, imperfect forms of creativity and highlight the importance of training on realistic, user-generated data. GUESSBENCH not only reveals nuanced gaps in model performance but also shows potential for enhancing VLM capabilities through fine-tuning on reasoning traces for creativity sensemaking. We envision GUESSBENCH as a valuable resource in aligning VLMs more closely with the complex and inclusive landscape of human creative expression.

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

## LIMITATIONS

There are two primary limitations. First, due to disparities in technological development across countries and regions, the majority of players in the "Guess the Build" game on the Hypixel server are from more developed areas. As a result, the data we collect lacks balanced geographical representation at the global level. Second, in our dynamic task setting, current VLMs do not yet support directly answering questions based on video input. Therefore, we simulate the dynamic process using three static images accompanied by corresponding hints. We anticipate that both limitations can be addressed in the future as global technological infrastructure improves and any-to-any models become available.

## ETHICS STATEMENT

Due to the presence of social and cultural biases (Naous et al., 2024), VLMs tend to exhibit higher accuracy when predicting certain concepts over others, and their interpretations often align more closely with Western cultural contexts. For example, as shown in Figure 7, the Minecraft build represents the logo of China Post. While players familiar with this symbol can easily identify it, GPT-4o describes it as "fast food". This example illustrates that VLMs exhibit a clear cultural bias in their understanding of creative content.

## REPRODUCIBILITY STATEMENT

We release the data of GUESSBENCH at the anonymous link `https://anonymous.4open.science/r/GuessBench-E863/` , along with the code for obtaining model outputs on GUESSBENCH and computing the corresponding accuracy. In addition, the Supplementary Material includes the code and the first 40 samples of GUESSBENCH. Furthermore, §2 provides a detailed description of the design of GUESSBENCH, including the task setting (§2.1), data curation (§2.2), and evaluation metrics (§2.3).

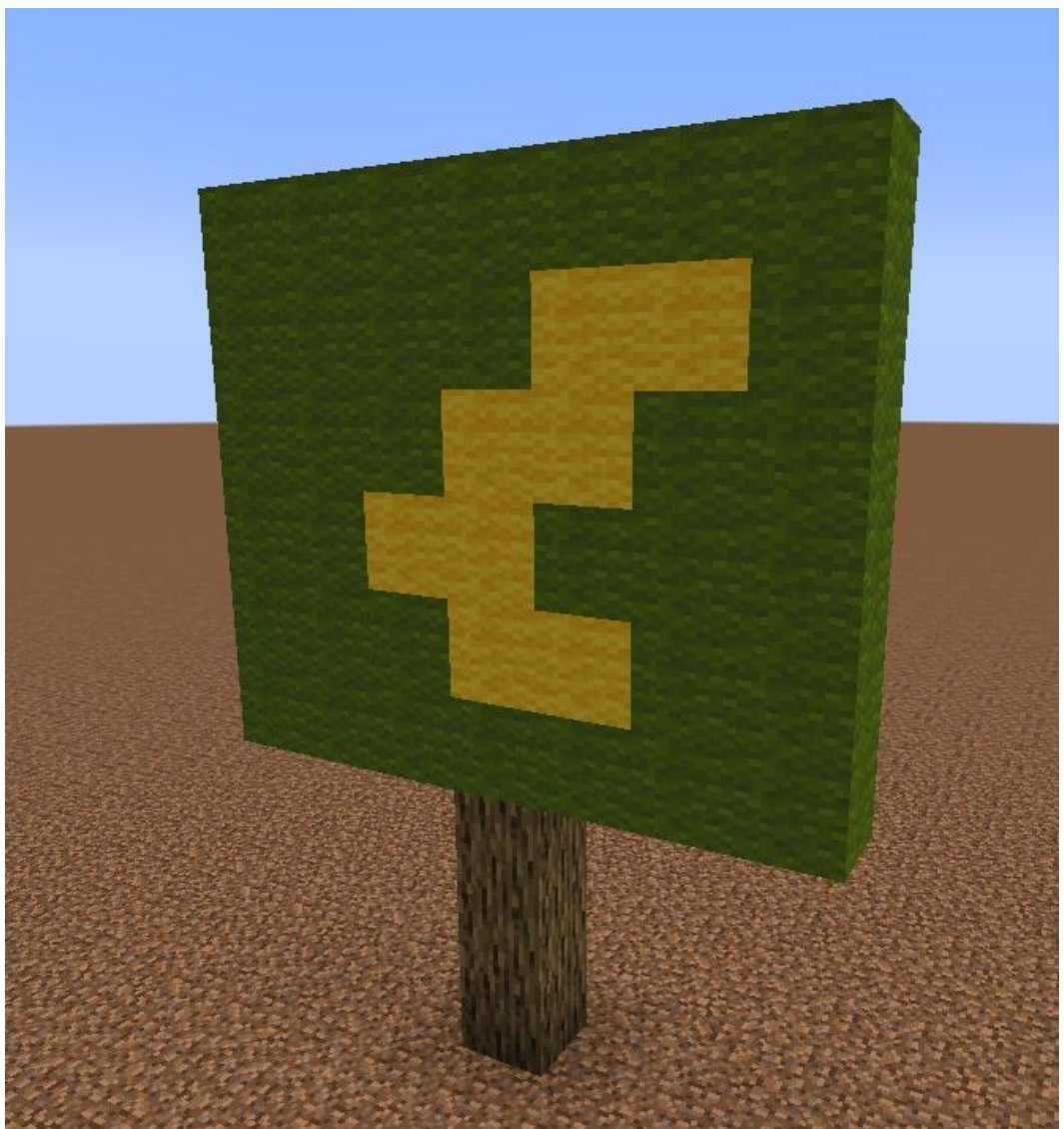

Figure 7: A Minecraft build depicting the China Post logo.

## A EXPERIMENT DETAILS

### A.1 INSTRUCTIONS

Table 3 presents the default instructions used in GUESSBENCH. For the static task, where only a single guess is allowed, we use the *Initial Guess Instruction* exclusively. For the dynamic task, the *Initial Guess Instruction* is used for the first attempt. If the response is incorrect, subsequent attempts incorporate the model's previous guess to construct the *Guess Refinement Instruction*.

### A.2 ANSWER EXTRACTION PROMPT

Table 4 presents the instruction provided to GPT-4o for extracting the predicted guess generated by the VLMs.

| Category | Detail |
|---|---|
| System | You are an AI designed to play the "Guess the Build" game from the Hypixel server in Minecraft. Your task is to analyze a player's build and accurately guess what it represents. The build is constructed using Minecraft blocks, and it can depict objects, animals, structures, abstract concepts, or any other recognizable entity.

Consider common themes, contextual clues, and typical constructions in Minecraft. If a build is unclear, make educated guesses based on possible interpretations. Your goal is to guess as accurately and efficiently as possible.

Avoid overly broad answers—be specific when possible. If multiple plausible answers exist, prioritize the most likely one based on common builds in the game. |
| Initial Guess | Look at the provided image of a Minecraft build and the corresponding hint. Based on its structure, shape, materials, and the given hint, determine what it represents. Consider common objects, animals, buildings, abstract concepts, or any other recognizable entities typically built in Minecraft.

You can use chain of thought reasoning to arrive at the best possible guess. |
| Guess Refinement | Your previous guess is: *previous_guess*. This guess is incorrect. Now, you will be provided with a more complete version of the build along with additional hints. Analyze the new information carefully and make a revised guess based on the updated build and hints.

You can use chain of thought reasoning to improve your guess. |

Table 3: Illustrations of Instructions. The table presents the default instructions used for evaluating GUESSBENCH. During evaluation, either the *Initial Guess Instruction* or the *Guess Refinement Instruction* is concatenated with the corresponding hint and then provided as input to the VLM.

| Instruction |
|---|
| I will give you a language model's response in a 'Guess the Build' game. Identify and extract the model's guessed answer. Output it in the format: 'Answer: [extracted answer]'. If no answer is given in the response, output 'No Answering'.
Language model's response: |

Table 4: Illustrations of the instructions used for guess extraction. The response from the VLM is appended directly after these instructions, and the combined input is then fed into GPT-4o to extract the predicted guess.

### A.3 PROMPT FOR DIFFERENT REASONING APPROACHES

Table 5 presents the core instructions for the different reasoning approaches. For the Self-Consistency method, no specific instruction is provided; instead, the VLM is prompted three times, and the final answer is determined via majority voting.

### A.4 MODEL HYPERPARAMETERS

Table 6 presents the detailed parameter configurations for the models discussed in (§3.1). Parameters not explicitly specified follow their default settings. Notably, for the Self-Consistency experiments described in (§3.2), we allow sampling and do not manually set the temperature or top_p parameters, instead relying on their default values. This design choice is intended to promote response diversity during generation, thereby ensuring that the results obtained through majority voting are meaningful.

### A.5 IMPLEMENTATION DETAILS FOR FINE-TUNING

We use the Unsloth framework (Daniel Han & team, 2023) and apply a consistent set of hyperparameters for the three fine-tuning settings: without tuning, synthetic tuning, and mixed tuning, as

| Approach | Detail |
|---|---|
| Without CoT | You are not allowed to use chain-of-thought reasoning. You must output the final answer directly. |
| One-shot | I will give you an example to help you guess the build. Here is an example:

¡Image¿
Hint_3: The answer format is as follows:
   l     s        k
- - - - -  - -  - - - -
This means the answer consists of 3 words. The 1st word has 5 letters. The 2nd word has 2 letters. The 3rd word has 4 letters. The 1st word's 2nd letter is l. The 1st word's 5th letter is s. The 3rd word's 4th letter is k.
Answer: The build represents Glass of Milk. |
| Self-Refine | Here is your response: *previousresponse*
Please review your answer to check if it aligns with the given hint. If it does, you MUST ONLY output "$Well done!$" Otherwise, provide your improved guess. |
| Image Retrieval | Previously, your guess was: *previous_guess*
Now, I have searched for a real image of *previous_guess* from Google Images and attached it above. Please compare the provided Minecraft build with the real image of your guess.
If they match, you MUST ONLY output: "$Well done!$". Otherwise, provide your improved guess. |

Table 5: Illustrations of the instructions used for different reasoning approaches. For each reasoning approach, we present the core instruction to provide a more intuitive and accessible understanding.

| Model | Generation Setup |
|---|---|
| GPT-4o | model = `gpt-4o-2024-11-20`, temperature = 0.0, top_p=1.0, max_tokens = 1000 |
| Gemini-2.0-Flash | model = `gemini-2.0-flash-001`, temperature = 0.0, top_p=1.0, max_output_tokens=1000 |
| GPT-4o-mini | model = `gpt-4o-mini-2024-07-18`, temperature = 0.0, top_p=1.0, max_tokens = 1000 |
| InternVL2.5-MPO | max_new_tokens = 1000, do_sample = False |
| Qwen2.5VL-72B | max_new_tokens = 1000, do_sample = False |
| MiniCPM-V2.6 | max_new_tokens = 1000, sampling = False |

Table 6: Generating parameters for VLMs.

shown in Table 7. The fine-tuning process is performed using a single NVIDIA A100-SXM4-40GB GPU.

## A.6   PERFORMANCE OF SELECTED MODELS ON EXISTING BENCHMARKS

We obtained the accuracy of all selected models on the four existing datasets from the OpenVLM Leaderboard[5]. To ensure consistency with the evaluation metric used in GUESSBENCH, which is the average accuracy per question, we selected the *Overall Accuracy* metric for the AI2D, MMBench V1.1 Test (EN), and MathVista benchmarks. For HallusionBench, we instead used *aAcc*, which represents the overall accuracy across all atomic questions.

---

[5]https://huggingface.co/spaces/opencompass/open_vlm_leaderboard

| Category | Detail |
|---|---|
| Model | model_name = "unsloth/Qwen2-VL-7B-Instruct", load_in_4bit = True |
| LoRA | r = 16, lora_alpha = 16, lora_dropout = 0, random_state = 3407, |
| Fine-tuning | per_device_train_batch_size = 2, gradient_accumulation_steps = 4, warmup_steps = 5, max_steps = 30, learning_rate = 2e-4, optim = "adamw_8bit", weight_decay = 0.01, lr_scheduler_type = "linear", seed = 3407, max_seq_length = 2048, |

Table 7: Fine-tuning Configuration and Hyperparameters.

| Dataset | GPT-4o | Gemini-2.0-Flash | InternVL2.5-78B-MPO | Qwen2.5VL-72B | MiniCPM-V2.6 |
|---|---|---|---|---|---|
| AI2D | 84.9 | 83.1 | 89.2 | 88.5 | 82.1 |
| MMBench V1.1 Test(EN) | 84.8 | 70.4 | 87.9 | 88.3 | 79.0 |
| MathVista | 60.0 | 70.4 | 76.6 | 74.2 | 60.8 |
| HallusionBench | 71.4 | 72.0 | 73.3 | 71.9 | 65.0 |

Table 8: The detailed accuracy of all models on the four existing datasets.

## A.7 LEVERAGING GUESSBENCH BEYOND EVALUATION

Under the static setting, we randomly sample 150 Minecraft build sets from the questions correctly answered by GPT-4o as the training set. From the remaining 350 build sets, we randomly select 240 as the test set. We then fine-tune Qwen2-VL-7B using supervised learning. As shown in Table 9, fine-tuning yields performance gains, indicating that GUESSBENCH serves not only as an evaluation benchmark but also as a valuable training resource for improving VLMs.

| Qwen2-VL-7B | Static | | | Dynamic #1 | | | Dynamic #2 | | | Dynamic #3 | | |
|---|---|---|---|---|---|---|---|---|---|---|---|---|
| | Short | Long | All | Short | Long | All | Short | Long | All | Short | Long | All |
| Original | 11.20 | 3.48 | 7.50 | 0 | 0.87 | 0.42 | 1.60 | 3.48 | 2.50 | 4.80 | 3.48 | 4.17 |
| Finetuned | 16.80 | 6.96 | 12.08 | 0.80 | 0.87 | 0.83 | 6.40 | 1.74 | 4.17 | 11.20 | 7.83 | 9.58 |

Table 9: Results of fine-tuning Qwen2-VL-7B on GUESSBENCH. Under both static and dynamic settings, fine-tuning consistently improves performance, demonstrating that GUESSBENCH can serve as a valuable training resource in addition to an evaluation benchmark.

# B SYCOPHANTIC BEHAVIOR IN VLMS UNDER CONTRADICTORY USER FEEDBACK

To investigate whether VLMs exhibit sycophantic behavior when confronted with contradictory user requests, we conduct a study focusing on two state-of-the-art models: GPT-4o and Gemini-2.0-Flash. Within the static evaluation setting, we select 100 instances from the Minecraft build sets that both models originally answered correctly. For each instance, after the VLM provides a correct response, we intentionally assert that its answer is incorrect, allowing us to observe whether the model revises its correct prediction and instead conforms to the user's false feedback.

We design three conditions to induce such behavior in VLMs. In the **Base** setting, we merely inform the VLM that its answer is wrong without providing any additional information. The **Random** setting builds upon the Base condition by additionally supplying a randomly chosen incorrect answer, sampled from the set of incorrect options in GUESSBENCH, and asserting it as correct. The **Similar** setting also builds upon the Base condition, but the misleading answer is semantically similar to the correct one, generated using GPT-4o, and presented as the correct answer.

The results reveal that GPT-4o maintains an accuracy of 6%, 15%, and 26% under the Base, Random, and Similar settings respectively. In contrast, Gemini-2.0-Flash achieves only 1%, 2%, and 4% accuracy across the same conditions. These findings suggest that Gemini-2.0-Flash is more susceptible to sycophantic behavior than GPT-4o.

Interestingly, VLMs are most likely to revise their initially correct answers in the Base setting, where no alternative answer is proposed, and are least likely to do so in the Similar setting, where a plausible but incorrect alternative is given. Further analysis of the model responses under contradictory user feedback reveals that in both the Random and Similar settings, the VLMs do not engage in any substantive evaluation of the user-supplied answer's plausibility. Instead, they often respond with generic phrases such as "reconsidering the build and the hint," immediately followed by a new, often incorrect, guess.

Paradoxically, as the misleading input becomes more semantically aligned with the correct answer from Base to Random to Similar, the models' accuracy increases. This trend runs counter to our hypothesis. If the models were capable of independent reasoning, misleading cues that closely resemble the correct answer would be more deceptive and thus reduce accuracy. Instead, our results suggest that VLMs fundamentally lack the ability to assess correctness or engage in independent thought. Rather, they merely generate text that appears syntactically plausible in response to user prompts.

## C   FLIPPING THE PARADIGM: EASIER TO GENERATE THAN DISCRIMINATE

Existing experiments show that generative tasks are generally more challenging for LMs than discriminative ones (Jiang et al., 2025). However, our findings reveal a surprising result: creatively generating a Minecraft build is significantly easier than identifying one, further underscoring the novelty and distinctiveness of GUESSBENCH.

In our generative experiment setup, we select the Gemini-2.0-Flash model and randomly sample 100 build sets from GUESSBENCH. For each sampled item, we manually input prompts into the Gemini playground[6], asking the model to generate an image of a Minecraft build based on a given answer, thereby reversing the question-answering direction in GUESSBENCH. For the discriminative counterpart, we adopt the *static task* from GUESSBENCH as our evaluation setting.

We evaluate performance using accuracy. A generation is considered correct only if it satisfies two criteria: it contains the specified entities, and it appears plausibly constructed within the Minecraft environment. Under this rigorous standard, Gemini-2.0-Flash achieves an accuracy of 80% on the generative task, which is substantially higher than its 40% accuracy on the corresponding discriminative task within GUESSBENCH.

This notable performance gap highlights a critical limitation in VLMs' ability to perform creative understanding in open-ended scenarios, a challenge that remains largely overlooked in prior work. GUESSBENCH not only exposes this limitation but also points to promising directions for future improvements in VLM capabilities.

## D   FROM FEEDBACK TO FORESIGHT: GPT-4O LEARNS IN CONTEXT LIKE A PLAYER

To more accurately simulate the gameplay of "Guess the Build" and evaluate the performance of VLMs in a realistic gaming environment, we divide a total of 500 questions into 100 sets, each containing 5 questions. This setup mirrors the actual game mechanics, where five questions are asked consecutively, and after each question, the VLM receives feedback indicating whether its guess is correct or incorrect. In our experiments, we adopt GPT-4o as the VLM under investigation.

As shown in Figure 8, answering five questions in succession leads to improved accuracy, with a more substantial performance gain observed in dynamic tasks. Specifically, accuracy increases by 4.8% in static tasks and by an average of 10.47% across three attempts in dynamic tasks. These results indicate that GPT-4o is able to extract more information from dynamic tasks and utilize prior interactions through in-context learning to enhance its responses to subsequent questions. Moreover, we observe that in dynamic tasks, the improvement in accuracy diminishes with each additional attempt. This suggests that the benefits of GPT-4o's in-context learning are more pronounced when dealing with more incomplete Minecraft builds and less informative hints.

---

[6]https://aistudio.google.com/prompts/new_chat

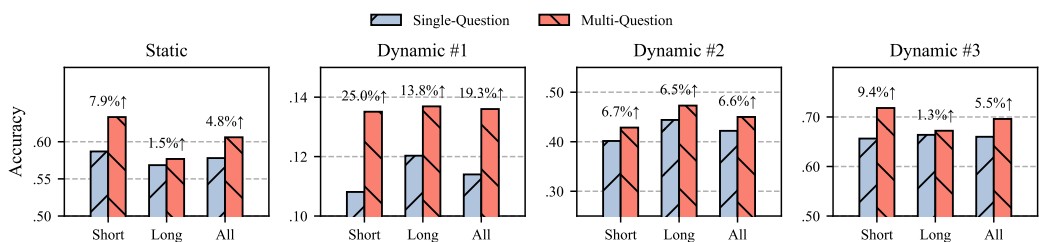

Figure 8: Evaluation results of Single-Question and Multi-Question settings. All denotes the overall accuracy for each task. Answering multiple questions sequentially within the same context improves response accuracy.

This experiment demonstrates that GPT-4o possesses strong in-context learning capabilities, enabling it to answer new questions more effectively by drawing on previous ones, similar to human players. It also highlights a promising direction for enhancing VLMs' ability to understand creativity in real-world contexts.

## E    REASONING RELIABILITY ANALYSIS

To evaluate whether VLMs can accurately explain their reasoning path when correctly identifying a Minecraft build, we conduct a manual analysis under the static task setting. Specifically, we randomly sample 50 responses from all correctly answered questions by GPT-4o and examine the reasoning path provided in each response. We find that 14% of the responses contain issues in their reasoning paths. Among these, 42.8% contain flawed reasoning (for example, inferring that a build resembles "a pit or a mine with ores inside" based on appearance, but ultimately identifying it as a "crate"), 28.6% involve errors in visual perception (such as stating that a table has four legs when the image clearly shows only three), and the remaining 28.6% result from inaccurate textual perception (for instance, misinterpreting the hint "_ l a _ _ _ _" as "l a"). These findings suggest that GPT-4o occasionally generates incorrect reasoning even when the final answer is accurate, underscoring its limitations in creativity understanding.

## F    EXAMPLES

### F.1    ILLUSTRATIVE EXAMPLES FOR STATIC AND DYNAMIC TASKS

For both the *static* and *dynamic* task settings, we selected the example "Graveyard" to illustrate the entire process within each task, including how GPT-4o receives the question, produces a response, and is subsequently evaluated for correctness. This example is intended to help readers gain a clearer understanding of the specific nature and structure of the two tasks.

### F.2    CASE STUDY: IMAGE RETRIEVAL REDUCES ANSWER ACCURACY

We present a case where the use of an **Image Retrieval** method unexpectedly reduces the accuracy of the model's prediction. Initially, GPT-4o correctly predicts the answer as *Modern Art*. However, since *Modern Art* encompasses a wide range of architectural styles and painting techniques, the image retrieval method returns an image of *Starry Night* by Vincent van Gogh. Although this painting falls under the category of *Modern Art*, it differs significantly from the original Minecraft build in visual appearance. As a result, GPT-4o alters its originally correct prediction. Subsequent image retrievals and comparisons fail to guide the model back to the correct answer.

## System Prompt

You are an AI designed to play the "Guess the Build" game from the Hypixel server in Minecraft. Your task is to analyze a player's build and accurately guess what it represents. The build is constructed using Minecraft blocks, and it can depict objects, animals, structures, abstract concepts, or any other recognizable entity.

Consider common themes, contextual clues, and typical constructions in Minecraft. If a build is unclear, make educated guesses based on possible interpretations. Your goal is to guess as accurately and efficiently as possible.

Avoid overly broad answers—be specific when possible. If multiple plausible answers exist, prioritize the most likely one based on common builds in the game.

## Dynamic Attempt #1

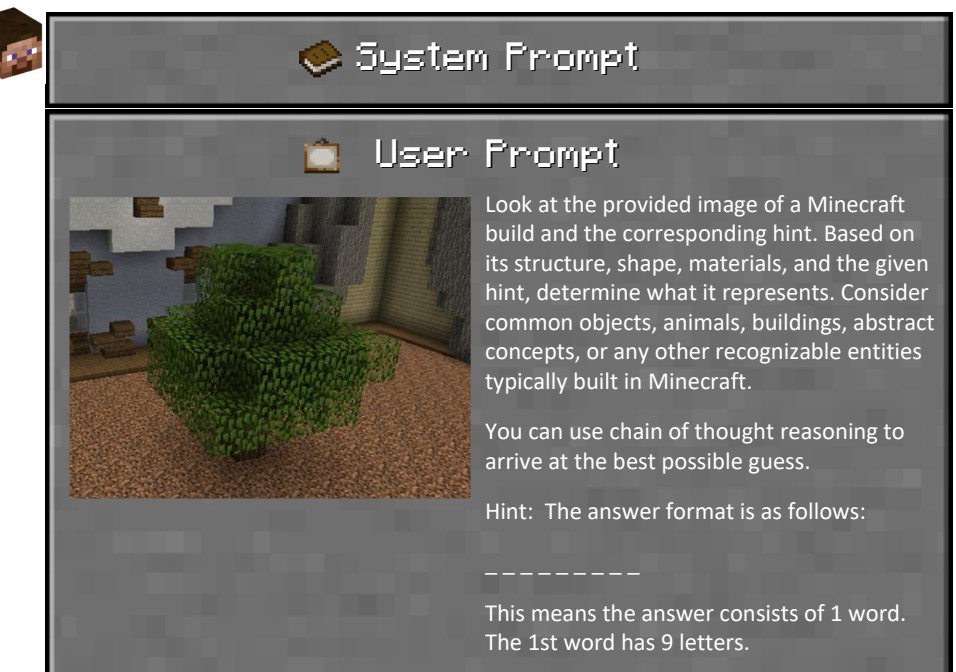

### System Prompt

### User Prompt

Look at the provided image of a Minecraft build and the corresponding hint. Based on its structure, shape, materials, and the given hint, determine what it represents. Consider common objects, animals, buildings, abstract concepts, or any other recognizable entities typically built in Minecraft.

You can use chain of thought reasoning to arrive at the best possible guess.

Hint: The answer format is as follows:

_ _ _ _ _ _ _ _ _

This means the answer consists of 1 word. The 1st word has 9 letters.

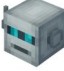

The build in the image appears to be a tree-like structure made primarily of leaves. Given the hint that the answer is a single word with 9 letters, and considering the shape and materials used, the most likely guess is "Christmas" followed by "tree" to form "Christmas tree." However, since the hint specifies a single word, the correct answer should be "Christmas."

Therefore, the answer is "Christmas."

## Incorrect Answer. Attempt Failed.

Figure 9: An example illustrates the first attempt in the dynamic task setting: GPT-4o responds based on an incomplete Minecraft build and a partial hint, but produces an incorrect answer.

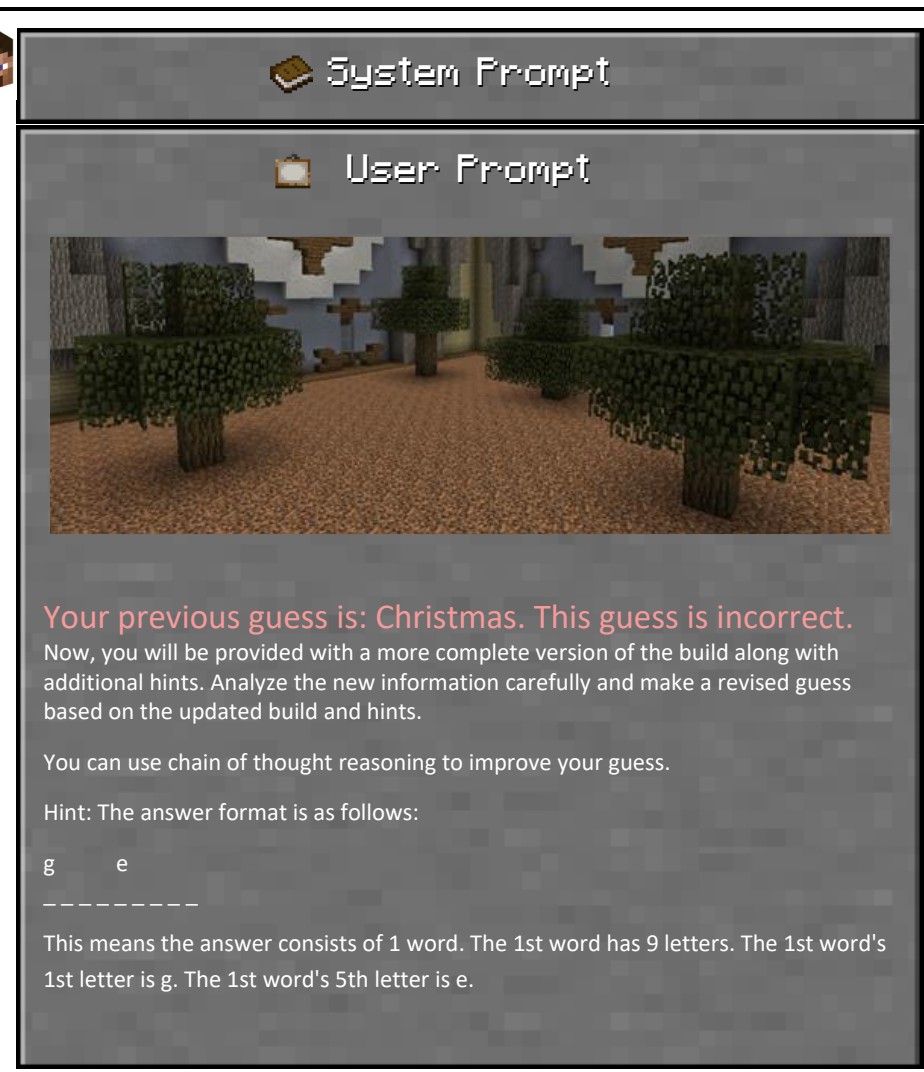

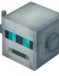

Figure 10: An example illustrates the second attempt in the dynamic task setting: GPT-4o, with access to the details of the first attempt, responds again based on a more complete Minecraft build and a more informative hint, but still produces an incorrect answer.

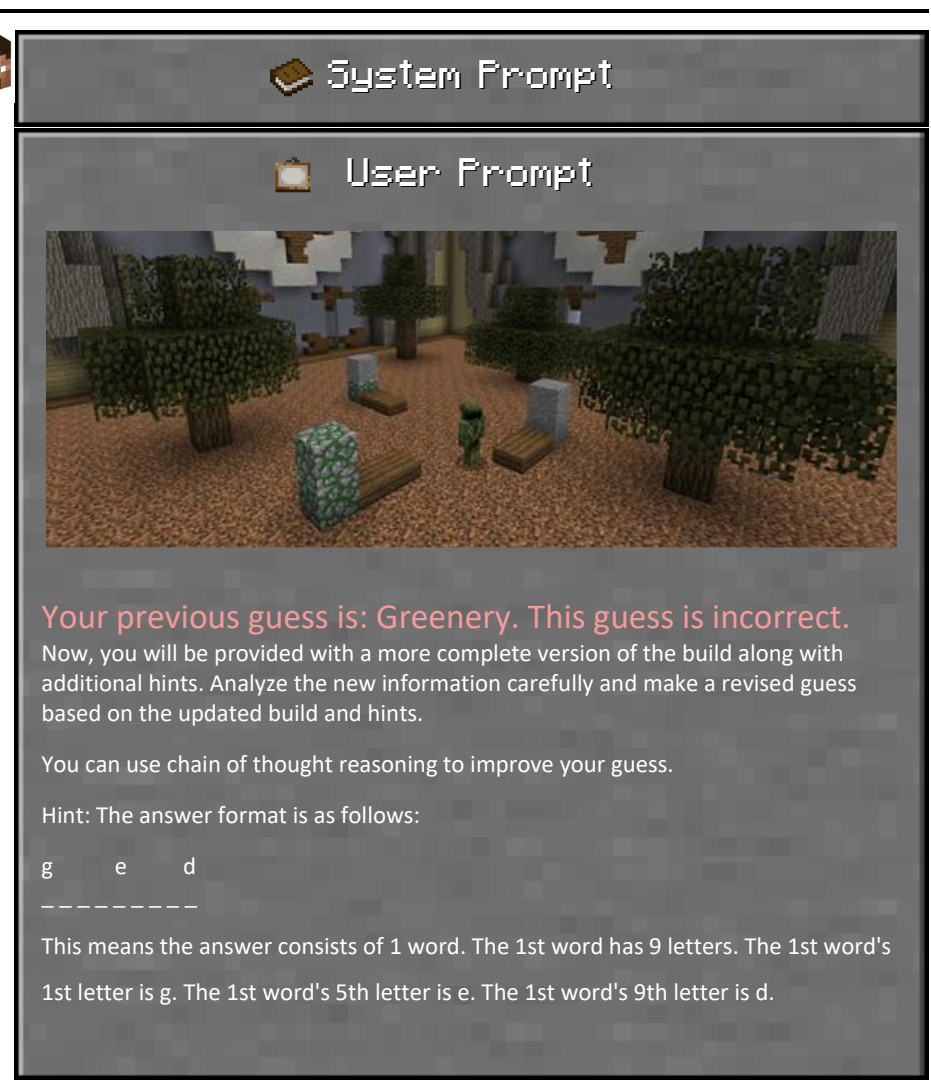

Figure 11: An example illustrates the third attempt in the dynamic task setting: GPT-4o, with access to the details of the previous two attempts, responds again based on the most complete Minecraft build and the most comprehensive hint, and produces a correct answer.

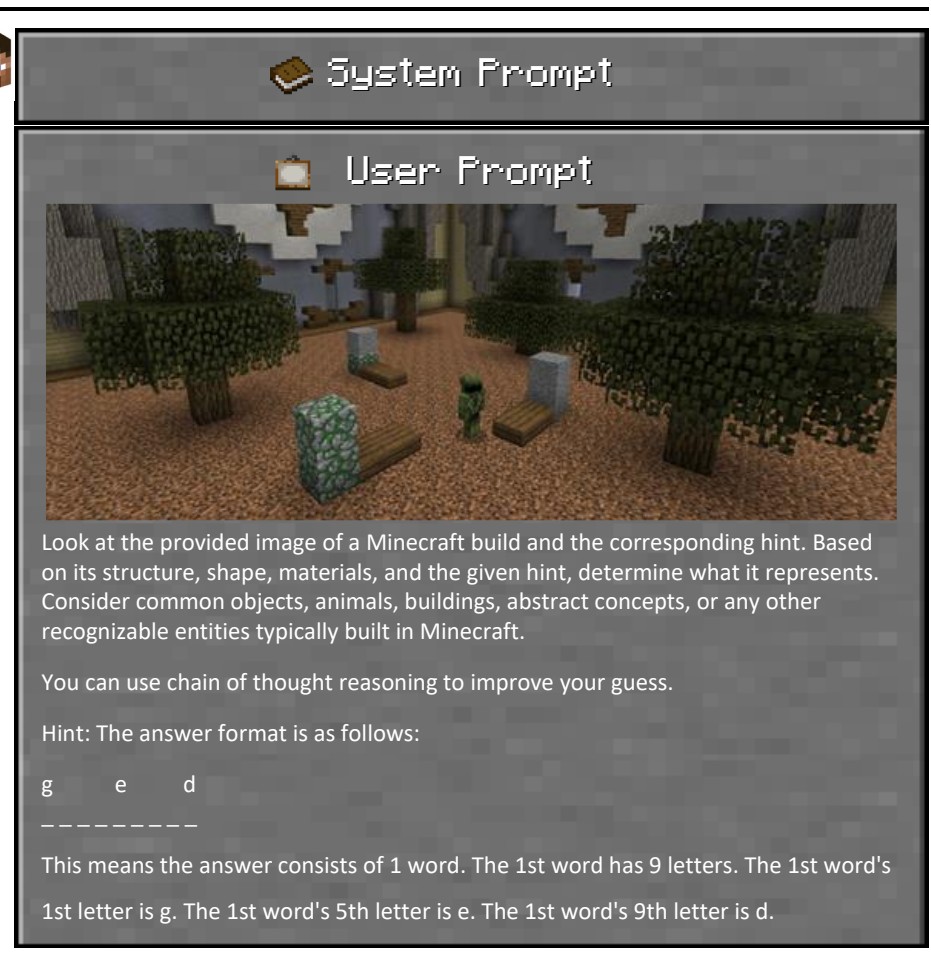

Figure 12: An example of the static task setting: GPT-4o, based on a complete Minecraft build and a full hint, makes a prediction and provides the correct answer.

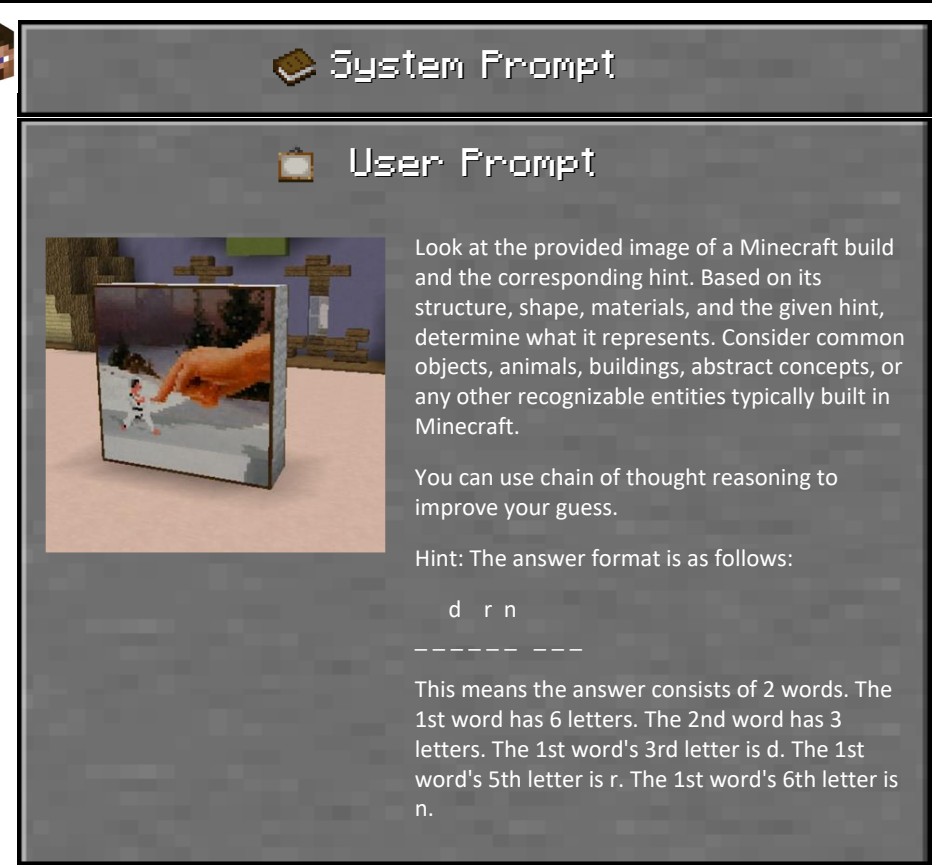

## Retrieval

**System Prompt**

**User Prompt**

Look at the provided image of a Minecraft build and the corresponding hint. Based on its structure, shape, materials, and the given hint, determine what it represents. Consider common objects, animals, buildings, abstract concepts, or any other recognizable entities typically built in Minecraft.

You can use chain of thought reasoning to improve your guess.

Hint: The answer format is as follows:

```
  d   r n
```

```
______ ___
```

This means the answer consists of 2 words. The 1st word has 6 letters. The 2nd word has 3 letters. The 1st word's 3rd letter is d. The 1st word's 5th letter is r. The 1st word's 6th letter is n.

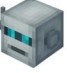

The image shows a Minecraft painting, which is a common decorative item in the game. The painting depicted is "The Pointer," which shows a hand pointing at a small figure. Given the hint and the structure of the answer, we need to find a phrase that fits the format:

- The first word has 6 letters, with the 3rd letter being 'd', the 5th letter being 'r', and the 6th letter being 'n'.
- The second word has 3 letters.

Considering the painting and the hint, the phrase "Modern Art" fits perfectly:

- M o d e r n (6 letters, with 'd' as the 3rd letter, 'r' as the 5th letter, and 'n' as the 6th letter)
- A r t (3 letters)

Therefore, the answer is "Modern Art."

(correct answer)

Figure 13: One example from the static task. Initially, GPT-4o provides the correct answer based on the Minecraft build and the accompanying hint.

## Refinement 1

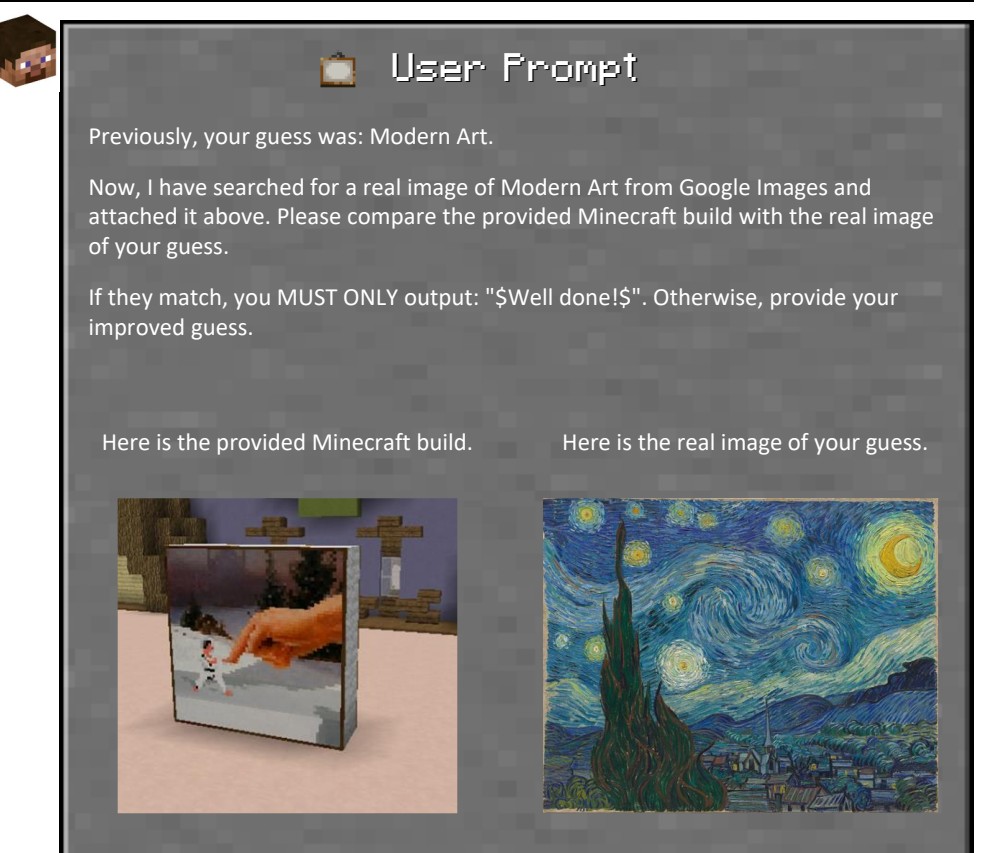

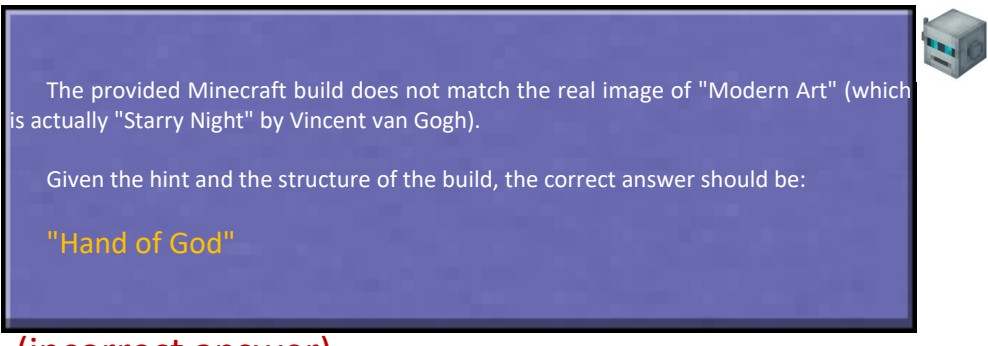

(incorrect answer)

Figure 14: Following the results of the first image retrieval, GPT-4o revises its initially accurate prediction after retrieving and comparing the images, resulting in an incorrect response. At this stage, GPT-4o remains in the refinement process and has not yet concluded, as it has not undergone evaluation.

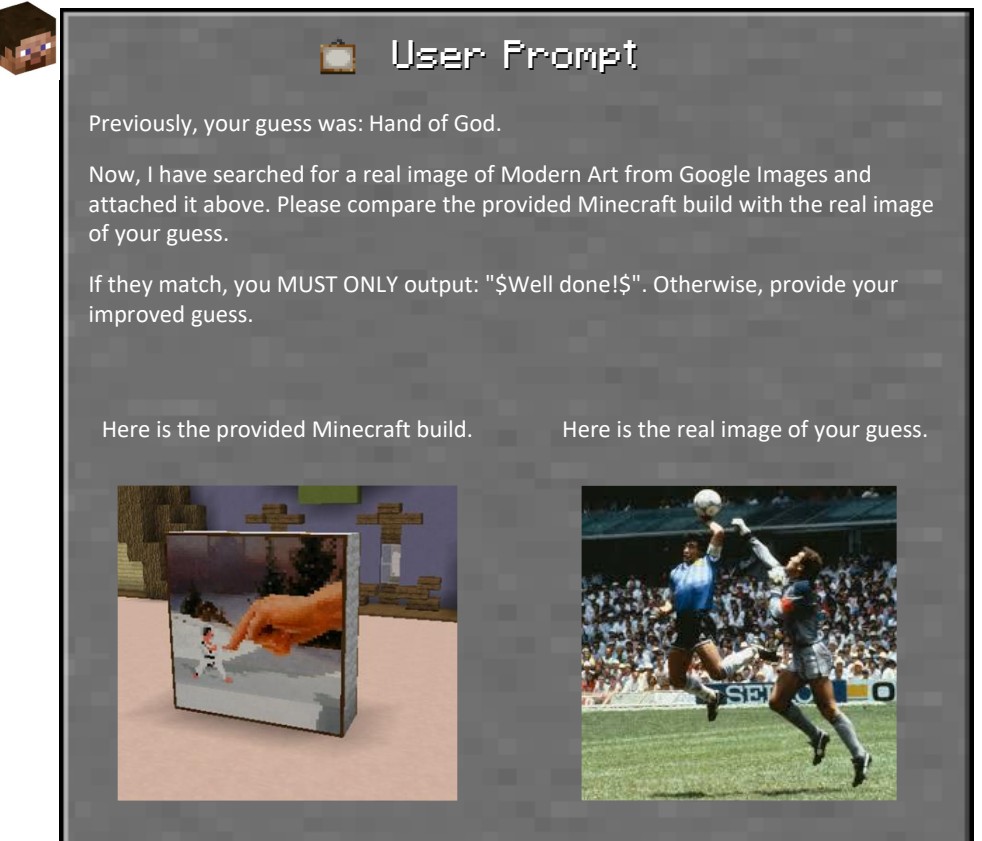

Figure 15: Subsequently, GPT-4o performs a second round of image retrieval. After retrieving and comparing the images, it again modifies its original prediction; however, the response remains incorrect. At this point, GPT-4o is still in the refinement process and has not yet reached completion, as it has not undergone final evaluation.

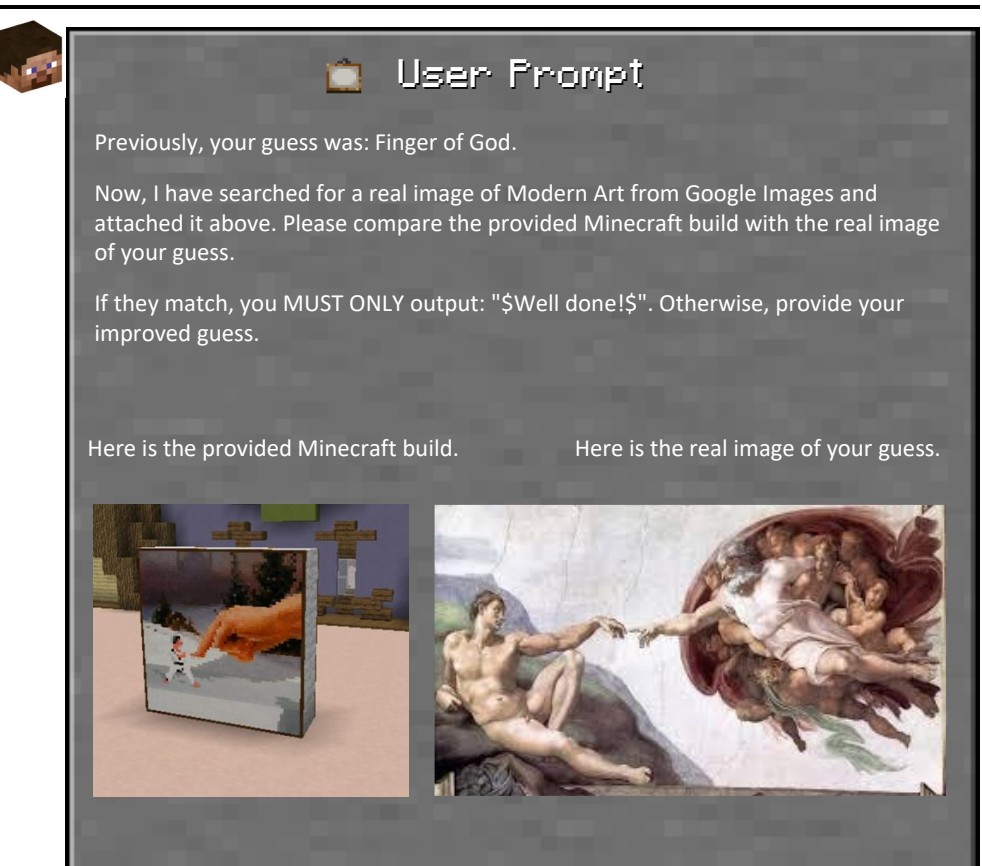

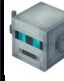

Figure 16: Finally, GPT-4o conducts a third round of image retrieval. After retrieving and comparing the images, it concludes that its original prediction is correct. It therefore submits "Finger of God", proposed in the previous round, as its final answer and proceeds to evaluation. However, this answer is incorrect.

