# OpenReview forum: "GuessBench: Sensemaking Multimodal Creativity in the Wild"
_ICLR.cc/2026/Conference — ICLR 2026 Conference Withdrawn Submission_

### Official Review · Reviewer_Yu1B · 2025-10-30

**Soundness:** 3
**Presentation:** 3
**Contribution:** 3
**Rating:** 6
**Confidence:** 3

**Summary:**

This paper introduces a novel benchmark, GUESSBENCH, designed to evaluate the "sensemaking" capabilities of Vision Language Models (VLMs) concerning human creativity. The authors argue that previous research has predominantly focused on evaluating elite-level creative generation, overlooking the need to understand the pervasive, noisy, and pluralistic "creativity in the wild" exhibited by ordinary users. The data is sourced from a popular Minecraft minigame, "Guess the Build," where VLMs act as guessers, needing to identify a concept from a player's build (either as a static image or a dynamic sequence) and accompanying language hints. The authors test a range of leading VLMs, finding that even GPT-4o faces significant challenges with this task and that a substantial performance gap exists between API and open-source models. The paper also includes a series of in-depth analyses on the impact of concept frequency, multilingual contexts, and cultural backgrounds on model performance.

**Strengths:**

1. Novelty and Importance of the Task: The paper's core contribution is shifting the evaluation focus from "creative generation" to "creative understanding." This is a critical and overlooked area. As VLMs are increasingly integrated into collaborative tools, their ability to understand user intent and "half-baked" creative ideas from non-expert users is paramount.
1. Ecological Validity of the Data Source: Using a real and popular online game like Minecraft as a data source is a significant advantage. It ensures that the visual data (player builds) truly reflects "creativity in the wild"—it is "noisy" and "pluralistic," not "sanitized" by researchers or scraped from professional art sites.
2. Static vs. Dynamic Settings: Providing both static (single image) and dynamic (image sequence) evaluation settings is a smart choice. The dynamic setting is particularly valuable as it simulates the process of progressive refinement in creativity, testing the model's ability to perform iterative reasoning and integrate new information.

**Weaknesses:**

1. Scale of the Dataset: The core dataset consists of 500 carefully curated build sets , with 424 unique answers (see Table 1 ). While the authors have clearly prioritized quality and manual curation, this is a relatively small number for a benchmark. This small scale might limit the statistical significance of some findings.
2. Simulation of "Dynamic" Task: The authors acknowledge this limitation, but it is a key one. The "dynamic" task is simulated using three static images from the build process.  This is a reasonable proxy, but it doesn't capture the full process of creation (e.g., seeing the builder place and remove blocks). This is more of a "progressive static image" task than a truly "dynamic" or video-based reasoning task. Three frames are a bit too few. Is there any ablations on number of frames?

3. Benchmark Comparison Set: In Section 5.1, the paper compares GUESSBENCH to MathVista and HallusionBench using the AutoBencher framework. While the quantitative metrics (Difficulty, Separability) are informative, the choice of benchmarks from entirely different domains (mathematical reasoning and hallucination detection) makes the comparison feel indirect.

**Questions:**

Human Evaluation Details: The paper mentions that human evaluation was conducted,  but provides no details. Who were the participants (e.g., expert Minecraft players, crowd-workers)? How many participants were there? What interface was used? These details are crucial for contextualizing the human baseline performance (e.g., 79.6% on static tasks ).

---

### Official Review · Reviewer_gDHW · 2025-11-01

**Soundness:** 3
**Presentation:** 3
**Contribution:** 2
**Rating:** 4
**Confidence:** 3

**Summary:**

This paper presents GuessBench, a benchmark evaluating multimodal “sensemaking” using human-created Minecraft builds. Models must infer intended concepts from static or dynamic build sequences with optional textual hints. The benchmark covers 2k examples and evaluates multiple VLMs (e.g., GPT-4o, Gemini, Claude). Results show large performance gaps and highlight biases and transfer patterns.

**Strengths:**

- Unique framing of creativity in the wild and sensemaking tasks.
- Rigorous evaluation across model families and reasoning modes.
- Thorough analysis of bias, difficulty, and transfer.
- Dataset and metrics well documented.

**Weaknesses:**

- Motivation for Minecraft domain could be further developed. Why not drawn or rendered images?
- The human baseline setup could be described in more detail.
- Contribution feels a bit more incremental than conceptual.

Minor comment:
- Some related work that could be relevant: Villareale, Jennifer, et al. "INNk: A multi-player game to deceive a neural network." Extended Abstracts of the 2020 Annual Symposium on Computer-Human Interaction in Play. 2020.
- Minor typo: “start-of-the-art” → “state-of-the-art.” in the abstract

**Questions:**

1. How were human baseline scores collected (number of participants)?
2. Could GuessBench be extended beyond Minecraft to test modality generalization?
3. How does it differ empirically from tests on creative drawing datasets?

---

### Official Review · Reviewer_fc7x · 2025-11-10

**Soundness:** 2
**Presentation:** 1
**Contribution:** 1
**Rating:** 2
**Confidence:** 4

**Summary:**

The paper introduces a small, targeted benchmark for evaluating vision–language models using images and prompts drawn from a Minecraft “pictionary”-style minigame. The stated aim is to measure “creativity in-the-wild” via guessing/building tasks. The dataset reportedly includes ~1.5k images and ~2k problems, with evaluation across 3 API models and 6 open-source models. The authors provide model configs and explore effects of fine-tuning, language variation, concept frequency, and multi-view inputs.



Presenting a novel benchmark dataset for vision language model evaluation.

Data is taken from a minecraft minigame where they basically play minecraft pictionary.

The data size doesn't seem overly significant at 1500 images and 2000 problems?

They keep using this "creativity in-the-wild" description of this minigame and I do feel like the game is not the "wild" like it's a videogame it is not like video taped games. The wild would be the real world in my mind.

They state that GuessBench introduces a quantitatively novel novel dataset compared to various VQA datasets and only references 2, let's double check that.

They said during data collection "we manually participate" how many is "we"? The Build collection feels a bit under developed in the writing.

Then in the ethical section they discuss things like they were crowd sourced? Who labelled this data set??

Validated with 3 API models and 6 open source ones - Great.

Good they give the models and their parameters

Table 2 you don't state what measurement is in the table in the caption please put that in. What is short and what is long again?

I do feel some of these results conclusions are a bit boring? Or too obvious? Like yes 4o is better than open source, yes iterative refinement is best.

Your fourth point is the interesting portion in my opinion.

Analysis section includes: Autobencher metrics, effect of fine tuning, effect of language variation, effect of concept freq, and effect of multiviews.

Section 5.1. your definition of difficulty makes it sound like you want the lowest difficulty score to be the best, maybe revise the wording

Figure order issue: Figure 4 comes before figrue 3 in the latex rendering, same with 5 and 6

grammatically there are a lot of "we"s and "this" and "it" used without good context. It would read clearer if you just stated what you are talking not expecting the reader to remember what you are referring to 2 or 3 sentences back.

In text figure 5 reference refers to the figure's left and right graphs where they are top and bottom oriented.

**Strengths:**

Clear motivation to probe compositional/creative reasoning under a constrained, reproducible environment.

Evaluation breadth is good (3 commercial + 6 OSS models) with parameters disclosed—this is appreciated for reproducibility.

Analysis sections (e.g., language variation, multiviews) are potentially useful diagnostics beyond a single leaderboard.

**Weaknesses:**

1. Overclaiming “in-the-wild.” Calling a controlled video-game environment “in-the-wild” feels overstated. “In-game creative behavior” or “player-generated scenes” would be more accurate and avoids suggesting real-world capture. Please recalibrate the framing.

2. Scale and positioning. At ~1.5k images/~2k problems, the benchmark is small by current VLM standards. That’s not disqualifying, but it argues for positioning this as a diagnostic suite rather than a general benchmark, and for strong leakage controls and confidence intervals.

3. Related work gap / novelty claim. The paper says the dataset is “quantitatively novel” vs VQA, but cites only two datasets. Given the extensive prior landscape (multiple VQA/VLM datasets spanning general, OCR-centric, knowledge-heavy, and instruction-following settings), this comparison needs to be far broader and more concrete (coverage, difficulty, artifacts avoided, diagnostic value).

4. Who labeled what? The data collection/labeling story is confusing. You write “we manually participate,” yet the ethics section refers to the process as if it is “crowdsourced” work. Please state clearly: (a) annotator counts and expertise, (b) instructions/guidelines, (c) inter-annotator agreement/adjudication, (d) compensation if crowdsourced, and (e) which parts were author-generated vs third-party. Dataset papers are expected to provide such documentation (e.g., datasheets / data statements).

Contamination & governance. Because Minecraft content is ubiquitous online and likely appears in pretraining corpora, a contamination check is needed (near-duplicate screens against common web corpora, date cut-offs, or time-based splits). Also clarify versioning (semver), license, hosting permanence, and update policy/leaderboard rules to prevent test-set overfitting over time. (ICLR’s Code of Ethics expects transparency around human data and data provenance.)

Metrics & reporting clarity:
- Table and figure captions lack essential detail for interpretation. For example, the caption for Table 2 does not specify the metric(s) or units reported, nor does it define the “short” versus “long” settings. Please revise all captions to state the exact metrics, units, evaluation setup (e.g., short vs. long), and any aggregation method (e.g., mean ± CI, number of runs).
- Report uncertainty: multiple seeds (and prompt seeds for LLMs), CIs/bootstraps, not just point estimates.
- Section 5.1: the “difficulty” definition reads as if lower is “better”; please reword to avoid ambiguity.

Analysis takeaways feel expected. Several conclusions are unsurprising (e.g., GPT-4o > OSS; iterative refinement best). That’s fine, but the paper would benefit from deeper why (error analysis, failure typology, slice-level effects tied to the benchmark’s intended construct).

Presentation/clarity issues.
- Figure order: Fig. 4 appears before Fig. 3; same with 5/6.
- In-text reference to “left/right” for Fig. 5, but the layout is top/bottom.
- Heavy use of “we/this/it” without referents—many sentences would read clearer if you restate the noun phrase.

**Questions:**

1. Please provide a thorough related-work comparison table situating this dataset among VQA/VLM benchmarks (axes: domain, size, annotation type, tasks, metrics, known artifacts). Related to Weakness 3 above.

2. Who labeled the data? How many annotators? What were the guidelines and IAA (e.g., κ/α), and how were disagreements resolved? Related to Weakness 4 above.

3. How did you check for pretraining contamination or near-duplicates? If likely, can you quantify its impact on scores?

4. What is the license, hosting plan, and versioning policy? Will there be a stable hidden test set or an eval server?

**Details Of Ethics Concerns:**

Please include a concise datasheet/data card covering motivation, composition, collection process, preprocessing, potential biases, intended/out-of-scope uses, and maintenance plan; and a clear statement on consent/ToS for any scraped materials. Also include IRB/ethics review status if human participants (players/annotators) were involved.

---

### Official Review · Reviewer_YF82 · 2025-11-11

**Soundness:** 3
**Presentation:** 3
**Contribution:** 3
**Rating:** 6
**Confidence:** 2

**Summary:**

The authors introduce GuessBench, a Minecraft "Guess the Build"-based benchmark for evaluating sensemaking with partial information of VLMs. The benchmark frames two tasks: static (one final build image + a hint) and dynamic (three progressively completed build images + progressively revealing hints). The authors evaluate 9 popular API/open VLMs plus several prompting-based reasoning methods (e.g., self-consistency, self-refine, image retrieval). The dataset contains 500 build sets and 1,500 images, with hints that reveal word length and a small number of letters over attempts; a natural-language version of the hint is also provided. Reported results show GPT-4o at 57.8% (static) and 66.0% (dynamic #3). The authors argue the benchmark is difficult, separates models well, and is novel relative to MathVista and HallusionBench (quantified via AutoBencher). They further explore transfer via lightweight fine-tuning and provide analyses on language resource disparities, view aggregation, sycophancy, and modality ablations. Overall, this is an interesting benchmark with careful data curation efforts. My concerns are also outlined below.

**Strengths:**

- **Originality:** The paper targets sensemaking creativity "in the wild" using a social game where non-expert builders convey concepts imperfectly. This is a fresh angle: decoding noisy, pluralistic, personalized human creativity. The two-stage static/dynamic design requiring guesses under evolving visual + symbolic hints is well thought out. Figure 1 illustrates the task clearly with partial-letter hints and successive build images.
- **Data curation efforts:** The dataset is described with concrete statistics. The hint design is explicit and seems reproducible. Ethics steps are documented. Decoding configs and prompts seem standardized.
- **Empirical analyses:** The paper evaluates 9 models (3 API + 6 open) and several prompt-based reasoning strategies; it also runs modality ablations, showing the task is mainly multimodal. The multi-view analysis is a nice touch (more angles don't trivially solve the problem), which underscores the benchmark's difficulty.
- **Significance:** AutoBencher metrics indicate GuessBench is harder and more separating than MathVista and HallusionBench, with a higher novelty score. This helps argue that the benchmark adds distinct signal beyond existing VLM tests. The language resource analysis shows performance tracks language prevalence; other probes of real interactions are also quite interesting.

**Weaknesses:**

- **What exactly is “creativity” here?** The paper motivates creativity in the wild and sensemaking creativity, but the operationalization reduces to guessing a noun/phrase under partial info. This aligns more naturally with VQA-style answer prediction and visual abductive reasoning under noisy inputs than with divergent creativity per se. To substantiate the “creativity” claim, please (i) define which creativity facets are intended (e.g., ambiguity resolution, imaginative reconstruction, robustness to imperfect stimuli), (ii) relate the task to established convergent benchmarks (e.g., visual abductive reasoning), and (iii) how the tasks go beyond standard noisy recognition with textual constraints.
- **Hint format may bottleneck models.** The underscore-plus-letters pattern seems tailored to letter-level reasoning, which many VLM stacks handle unreliably due to subword tokenization. While a natural-language hint is included, the evaluation still foregrounds letter-position alignment, which may favor models with specialized token handling over broader "creativity" skills. An ablation with purely qualitative clues (no blanks/letters), or graded natural-language clues, would better test sensemaking.
- **GPT-4o extraction.** The pipeline uses GPT-4o to extract the final guess from model outputs. Might be helpful to do a small-scale human-audit error rate for the extractor. Also, how do you handle synonymous or paraphrastic answers, etc.?
- **Potential pretraining overlap.** This might be difficult to quantify, but do the authors have a sense of potential leakage of Minecraft-like images in those VLMs' pretraining datasets? This kind of imagery seems ubiquitous, but it's fine if quantification is difficult. Alternatively, could you consider analyzing the performance of VLMs on subset of those posts after the training cutoff date?
- **Fine-tuning evidence is thin.** Gains on MultiChartQA/BLINK but drops on MathVista make the transfer story mixed (+ the gain is minimal on MultiChartQA). More careful analyses might help. As is, I'd treat FT results as suggestive.
- **Frontier coverage.** Results are solid, but I would have liked more recent advanced models for context, though I don't think this is critical for acceptance.

**Questions:**

See Weakness.

**Details Of Ethics Concerns:**

The authors outlined their steps to ensure ethical compliance in Sec 2.4. Seems good to me.

---

### Note · Authors · 2025-11-12

**Comment:**

We would like to thank the reviewers for their thoughtful comments and feedback.

**Withdrawal Confirmation:**

I have read and agree with the venue's withdrawal policy on behalf of myself and my co-authors.